# Back to BERT in 2026: ModernGENA as a Strong, Efficient Baseline for DNA Foundation Models

**Alena Aspidova[1], Yuri Kuratov[1,2], Artem Shadskiy[1,3], Mikhail Burtsev[4], Veniamin Fishman[1,3,5]**

[1]AXXX, Moscow, Russia
[2]Moscow Independent Research Institute of Artificial Intelligence, Moscow, Russia
[3]Sirius University, Sochi, Russia
[4]London Institute for Mathematical Sciences, London, UK
[5]Institute of Cytology and Genetics, Novosibirsk, Russia
`minja.fishman@gmail.com`

## Abstract

Recent progress in DNA language models has been increasingly driven by large and complex systems, which can obscure the impact of improvements to standard NLP architectures. In this work, we study whether and how a modernized BERT-style backbone (ModernBERT) can be adapted to genomic sequence modeling to improve computational efficiency, training stability, and long-context handling. Under controlled experimental settings, we benchmark efficiency across a range of sequence lengths and evaluate downstream performance on the Nucleotide Transformer benchmark. The resulting model, ModernGENA, achieves a strong efficiency–quality trade-off and ranks among the top-performing models in our evaluation suite. To support reproducibility and to provide a solid default reference point for future architectural work in genomics, we release the full implementation and configuration of ModernGENA as an open, reusable baseline.

## 1 Introduction

DNA encodes a vast amount of biologically meaningful information, including regulatory logic, evolutionary constraints, and functional signals, that we are still far from fully decoding. Deep learning offers a promising route to learn general-purpose representations directly from sequence.

Existing approaches in genomics broadly follow two paradigms. Task-specific supervised models are trained end-to-end for particular tasks, such as Enformer (Avsec et al., 2021), Borzoi (Linder et al., 2025) and AlphaGenome (Avsec et al., 2026). In contrast, DNA foundation models first learn general-purpose sequence representations via self-supervised pretraining and are then adapted to downstream applications. Motivated by the goal of a single reusable backbone that transfers across many genomic tasks, DNA foundation modeling has expanded across architectural families, including CNN-based models such as ConvNova (Bo et al., 2025), Transformer models such as DNABERT (Ji et al., 2021), GENA (Fishman et al., 2025), DNABERT-2 (Zhou et al., 2023), and Nucleotide Transformer (Dalla-Torre et al., 2025), and SSM-inspired methods such as Caduceus (Schiff et al., 2024). Recent progress increasingly relies on composite, multi-module systems that integrate representations across scales, such as GENERator (Wu et al., 2025), Evo2 (Brixi et al., 2025), and Nucleotide Transformer v3 (Boshar et al., 2025). While often effective, these designs tend to increase computational cost by simultaneously scaling sequence length, representational resolution, and model size, thereby raising both the memory footprint and runtime requirements for training and inference.

Amid this emphasis on scaling and architectural complexity, it is easy to overlook continued progress in standard architectures. Recent Transformer refinements improve computational efficiency, training stability, and long-context handling, as in ModernBERT (Warner et al., 2025). Here we evaluate how well these improvements transfer to the genomics setting. This yields a strong baseline for systematic comparison of future advances and provides a modernized Transformer module that can

be reused when building more efficient architectures, where Transformers often serve as a key component.

**Contributions.** We evaluate how modern Transformer refinements transfer to genomics. Specifically, we

1. adapt and evaluate modern Transformer refinements for DNA foundation modeling, following ModernBERT advances;

2. benchmark efficiency–quality trade-offs under controlled experimental settings;

3. release training & finetuning code, configuration, and pretrained models ModernGENA-base (135M) and ModernGENA-large (377M) as a strong baselines for future work.

## 2 RELATED WORK

Among contemporary encoder-style foundation models for DNA, three widely used baselines are DNABERT-2 (Zhou et al., 2023), GENA-LM (Fishman et al., 2025), and Nucleotide Transformer (Dalla-Torre et al., 2025). While all follow masked-language pretraining (MLM), they differ most clearly in positional mechanisms, attention efficiency choices, and tokenization.

**Tokenization:** NTv2 uses fixed k-mer tokens (6-mers), whereas DNABERT-2 and GENA-LM rely on variable-length BPE tokenization (Sennrich et al., 2016).

**Positional information:** NTv2 incorporates rotary positional embeddings (RoPE) (Su et al., 2024) and expands its token-level context window; GENA-LM uses absolute positional encodings, while DNABERT-2 replaces explicit positional embeddings with ALiBi attention biases (Press et al., 2021) to reduce reliance on a learned position table and improve length extrapolation.

**Attention efficiency:** DNABERT-2 explicitly integrates FlashAttention (Dao et al., 2022) to speed up and reduce memory use in self-attention. While DNABERT-2 provides a FlashAttention-based implementation, we were unable to enable it in our environment due to software incompatibilities: the installation procedure recommended in the official repository pins a Triton version that is incompatible with the released code. GENA-LM uses BERT-like full attention for shorter sequences and provides a specialized long-context variant with sparse attention (BigBird) (Zaheer et al., 2020). NTv2 retains full attention but modernizes the transformer block (including RoPE and more efficient MLP variants) to support a larger context window.

## 3 EXPERIMENTS

### 3.1 ARCHITECTURE

We instantiate two model sizes: **ModernGENA base** (135M parameters) and **ModernGENA large** (377M parameters). ModernBERT (Warner et al., 2025) serves as the backbone, which is an encoder-only Transformer modernized for stable training and high throughput on long sequences. The architecture incorporates the following key design choices:

- It scales better to long contexts in both representation quality and computational efficiency: it replaces absolute positional embeddings with RoPE (Su et al., 2024), and uses a hybrid attention pattern that alternates local sliding-window attention with global attention, with separate RoPE parameterizations for local and global layers.

- It improves optimization stability via a pre-norm block design (Xiong et al., 2020), an additional LayerNorm after the embedding layer, simplified normalization in the first attention block, and GeGLU in the FFN for a more expressive nonlinearity (Shazeer, 2020).

- It reduces unnecessary parameterization by disabling bias terms in most linear layers and in LayerNorm (except for the final linear layer).

- It accelerates training and inference through end-to-end unpadding (Zeng et al., 2022), variable-length kernel implementations (FlashAttention v3 (Shah et al., 2024) for global layers and v2 (Dao, 2023) for local layers), and compilation of compatible modules with torch.compile (Ansel et al., 2024).

Further details on the model architecture are provided in Appendix A.

## 3.2 TRAINING

### 3.2.1 GENOMIC DATASETS AND PREPROCESSING

The training corpus comprises all vertebrate species with available genome and transcript annotations in the NCBI RefSeq database as of December 2024 (443 assemblies), totaling 353,574,093,776 bp; the full list of assemblies is provided in Appendix D.

We focus on promoter regions as training intervals, since regulatory elements are enriched around transcription start sites (TSSs), providing a more informative signal for gene regulation tasks. For each gene and pseudogene, we extract a $[-16\,\mathrm{kbp}, +8\,\mathrm{kbp}]$ window around each unique TSS. Overlapping windows are merged into non-overlapping regions using BEDTools. For each resulting region, we include both strands (forward and reverse complement). Sequences containing ambiguous nucleotides (symbols other than A/C/G/T) are excluded.

### 3.2.2 TRAIN/VALIDATION SPLIT

The data are partitioned at the level of whole chromosomes. For all assemblies except human (GCF_000001405.40), the validation set accounts for $\sim$10% of the total genome length, with the remaining $\sim$90% used for training. For the human assembly, chromosomes 8, 20, and 21 are held out for validation, and all remaining chromosomes are used for training.

### 3.2.3 TOKENIZATION

Use a 32k BPE vocabulary over the A/T/G/C/N symbols, with special tokens [CLS], [SEP], [PAD], [UNK], and [MASK] following (Fishman et al., 2025). As a preprocessing step, long runs of N are collapsed into a single token.

### 3.2.4 TRAINING SETTINGS

The model was trained on 8 NVIDIA A100 80GB GPUs with a global batch size of 4096 sequences. To improve robustness to variable input lengths, we used dynamic sequence packing with sequence lengths sampled uniformly from 10 to 1024 tokens (average length $\approx$ 700 tokens). We optimized using decoupled AdamW with learning rate $4 \times 10^{-4}$, $\beta_1 = 0.9$, $\beta_2 = 0.98$, $\epsilon = 10^{-6}$, and weight decay $10^{-5}$. Weight decay was not applied to bias parameters or normalization layers. We used a warmup–stable schedule in token space: the learning rate was linearly warmed up over the first $3 \times 10^9$ tokens and then held constant for the remainder of training.

ModernGENA base was trained for 158 epochs and processed 1,510B tokens in total. ModernGENA large has been trained for 138 epochs, corresponding to 1,320B processed tokens. On 8×A100-80GB, the observed pretraining throughput was 1.33M tokens/s for ModernGENA base and 0.486M tokens/s for ModernGENA large.

## 3.3 COMPARISON GROUP

To isolate the effect of modern Transformer-block refinements on performance and computational efficiency, the evaluation focuses on a *primary baseline set* of encoder-only Transformer models with fewer than 500M parameters: DNABERT-2, NTv2, and GENA-LM. Since NTv2 is available in multiple configurations spanning a wide range of parameter counts, NTv2 (100M) and NTv2 (250M) are selected to match the size of the ModernGENA models and ensure a fair comparison. When directly comparable metrics are available for other models with different architectures and model sizes, they are included as additional reference points, but they are not part of the primary baseline set.

## 3.4 INFERENCE EFFICIENCY EVALUATION

Inference efficiency is measured as throughput (tokens/s) on fixed-length sequences using an NVIDIA A100 (80 GB) GPU. The evaluation considers the *primary baseline set* of encoder-only

Transformer models. For each sequence length $L$, the maximum batch size that fits in memory is determined via exponential growth until an error, followed by binary search. Using this batch size, throughput is averaged over 10 timing runs per model (Appendix B).

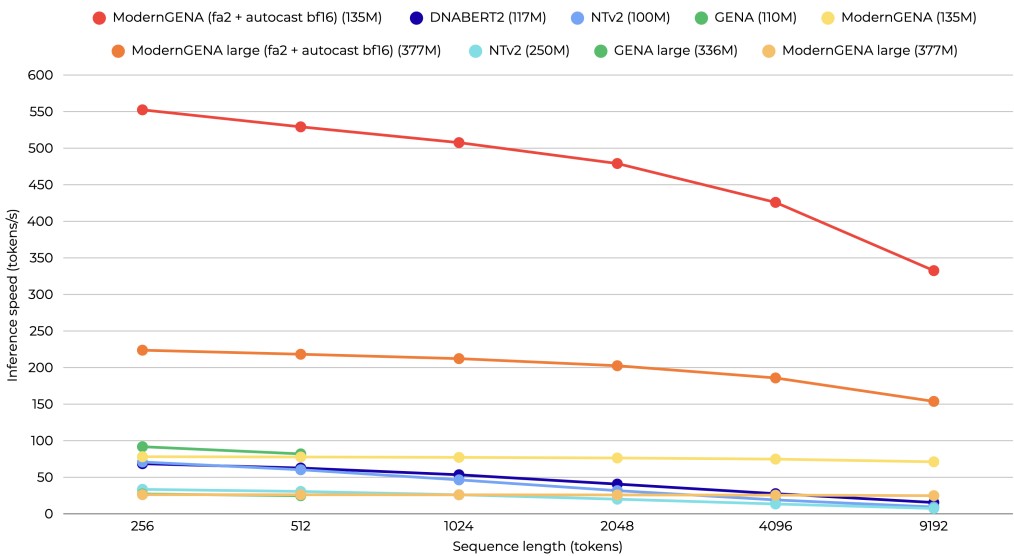

Figure 1: **Inference efficiency on an NVIDIA A100 (80 GB).** Models from the *primary baseline set* are benchmarked. Throughput is averaged over 10 timing runs per model.

Across these settings, ModernGENA achieves slightly higher throughput under the standard inference configuration and substantially higher throughput with FlashAttention 2, for both ModernGENA and ModernGENA Large (Figure 1).

### 3.5 RESULTS ON NT BENCHMARK

To enable fair comparison with prior architectures, ModernGENA is evaluated on the NT benchmark (Dalla-Torre et al., 2025), which consists of 18 tasks. ModernGENA is fine-tuned on each task without reverse-complement (RC) augmentation or conjoining; full training details are provided in Appendix C. Performance is reported as Matthews correlation coefficient (MCC) under 10-fold cross-validation. For each task, the rank of each model is computed and averaged across tasks. Results for the other models are taken from (Dalla-Torre et al., 2025; Wu et al., 2025).

ModernGENA (135M) achieves competitive performance across a broad range of tasks. Compared to GENA, it improves results on most benchmarks

Table 1: **NT Benchmark results (*primary baseline set*).** 10-fold cross-validation, reported as $100 \times$ MCC in the format *mean* $\pm$ *std* across folds. The final row reports each model's average rank across tasks.

| Task | DNABERT-2 (117M) | NTv2 (100M) | NTv2 (250M) | GENA (110M) | ModernGENA (135M) |
|---|---|---|---|---|---|
| H2AFZ | $49.0 \pm 1.3$ | $49.2 \pm 1.2$ | $51.3 \pm 1.7$ | $49.5 \pm 1.1$ | $52.2 \pm 0.7$ |
| H3K27ac | $49.1 \pm 1.0$ | $48.7 \pm 1.6$ | $49.7 \pm 1.4$ | $49.8 \pm 1.7$ | $51.2 \pm 1.3$ |
| H3K27me3 | $59.9 \pm 1.0$ | $59.5 \pm 1.3$ | $60.0 \pm 0.9$ | $60.0 \pm 0.7$ | $60.8 \pm 0.4$ |
| H3K36me3 | $63.7 \pm 0.7$ | $61.7 \pm 0.6$ | $63.6 \pm 1.2$ | $62.3 \pm 0.9$ | $63.8 \pm 0.9$ |
| H3K4me1 | $49.0 \pm 0.8$ | $48.5 \pm 1.1$ | $49.1 \pm 0.6$ | $48.6 \pm 0.9$ | $49.3 \pm 1.0$ |
| H3K4me2 | $55.8 \pm 1.3$ | $55.1 \pm 1.0$ | $57.0 \pm 0.9$ | $55.4 \pm 0.8$ | $57.5 \pm 1.1$ |
| H3K4me3 | $64.6 \pm 0.8$ | $63.3 \pm 1.5$ | $64.0 \pm 0.9$ | $65.7 \pm 1.2$ | $64.9 \pm 1.4$ |
| H3K9ac | $56.4 \pm 1.3$ | $53.8 \pm 1.5$ | $56.5 \pm 2.1$ | $54.3 \pm 1.0$ | $57.3 \pm 0.8$ |
| H3K9me3 | $44.3 \pm 2.5$ | $44.5 \pm 1.7$ | $46.7 \pm 1.6$ | $49.1 \pm 1.4$ | $48.2 \pm 1.8$ |
| H4K20me1 | $65.5 \pm 1.1$ | $64.8 \pm 0.8$ | $65.2 \pm 0.6$ | $65.6 \pm 0.9$ | $66.9 \pm 0.6$ |
| Enhancer | $51.7 \pm 1.1$ | $50.7 \pm 0.9$ | $52.5 \pm 1.0$ | $53.6 \pm 0.7$ | $54.5 \pm 0.6$ |
| Enhancer type | $47.6 \pm 0.9$ | $46.5 \pm 0.9$ | $49.2 \pm 1.0$ | $49.3 \pm 0.7$ | $51.2 \pm 1.3$ |
| Promoter all | $75.4 \pm 0.9$ | $75.3 \pm 0.5$ | $77.4 \pm 1.3$ | $74.1 \pm 1.1$ | $77.4 \pm 0.6$ |
| Promoter non-TATA | $76.9 \pm 0.9$ | $76.6 \pm 1.4$ | $78.5 \pm 1.0$ | $75.7 \pm 0.8$ | $77.9 \pm 0.9$ |
| Promoter TATA | $78.4 \pm 3.6$ | $82.6 \pm 1.9$ | $87.0 \pm 1.9$ | $80.7 \pm 4.9$ | $90.6 \pm 2.0$ |
| Splice acceptor | $83.7 \pm 0.6$ | $94.7 \pm 0.3$ | $95.0 \pm 0.8$ | $80.5 \pm 0.9$ | $85.1 \pm 0.3$ |
| Splice site all | $85.5 \pm 0.5$ | $96.0 \pm 0.3$ | $96.5 \pm 0.3$ | $82.3 \pm 0.8$ | $87.6 \pm 0.3$ |
| Splice donor | $86.1 \pm 0.4$ | $94.7 \pm 0.8$ | $96.7 \pm 0.4$ | $81.9 \pm 0.7$ | $86.8 \pm 0.6$ |
| **Average rank** | **3.67** | **4.17** | **2.22** | **3.33** | **1.50** |

and ranks among the top-performing models. By average rank, ModernGENA places first within the *primary baseline set* (Table 1) and second overall when including additional reference models across different architectures and model sizes (Table 2), with GENERator (1.2B) ranking first.

Table 2: **NT Benchmark results (all models).** 10-fold cross-validation, reported as $100 \times$MCC in the format *mean ± std* across folds. The final row reports each model's average rank across tasks.

| Task | ModernGENA (135M) | NTv2 (500M) | NT-multi (2.5B) | Enformer (252M) | HyenaDNA (55M) | Caduceus-Ph (8M) | Caduceus-PS (8M) | GROVER (87M) | GENERator (1.2B) | GENERator-All (1.2B) |
|---|---|---|---|---|---|---|---|---|---|---|
| H2AFZ | 52.2 ± 0.7 | 52.4 ± 0.8 | 50.3 ± 1.0 | 52.2 ± 1.9 | 45.5 ± 1.5 | 41.7 ± 1.6 | 50.1 ± 1.3 | 50.9 ± 1.3 | 52.9 ± 0.9 | 50.6 ± 1.9 |
| H3K27ac | 51.2 ± 1.3 | 48.8 ± 1.3 | 48.1 ± 2.0 | 52.0 ± 1.5 | 42.3 ± 1.7 | 46.4 ± 1.8 | 46.4 ± 2.2 | 48.9 ± 2.3 | 54.6 ± 1.5 | 49.6 ± 1.4 |
| H3K27me3 | 60.8 ± 0.4 | 61.0 ± 0.6 | 59.3 ± 1.6 | 55.2 ± 0.7 | 54.1 ± 1.8 | 54.7 ± 1.0 | 56.1 ± 3.6 | 60.0 ± 0.8 | 61.9 ± 0.8 | 59.0 ± 1.4 |
| H3K36me3 | 63.8 ± 0.9 | 63.3 ± 1.5 | 63.5 ± 1.6 | 56.7 ± 1.7 | 54.3 ± 1.0 | 54.3 ± 0.9 | 60.2 ± 0.8 | 58.5 ± 0.8 | 65.0 ± 0.6 | 62.1 ± 1.3 |
| H3K4me1 | 49.3 ± 1.0 | 49.0 ± 1.7 | 48.1 ± 1.2 | 50.4 ± 2.1 | 43.0 ± 1.4 | 41.1 ± 1.2 | 43.4 ± 3.0 | 46.8 ± 1.1 | 50.4 ± 1.0 | 49.0 ± 1.6 |
| H3K4me2 | 57.5 ± 1.1 | 55.2 ± 1.3 | 55.2 ± 2.2 | 62.6 ± 1.5 | 52.1 ± 2.4 | 48.0 ± 1.3 | 52.6 ± 3.5 | 55.8 ± 1.2 | 60.7 ± 1.0 | 56.9 ± 1.2 |
| H3K4me3 | 64.9 ± 1.4 | 62.7 ± 2.0 | 61.8 ± 1.5 | 63.5 ± 1.9 | 59.6 ± 1.5 | 58.8 ± 2.0 | 61.1 ± 1.5 | 63.4 ± 1.1 | 65.3 ± 0.8 | 62.8 ± 1.8 |
| H3K9ac | 57.3 ± 0.8 | 55.1 ± 1.6 | 52.7 ± 1.7 | 59.3 ± 2.0 | 48.4 ± 2.2 | 51.4 ± 1.4 | 51.8 ± 1.8 | 53.1 ± 1.4 | 57.0 ± 1.7 | 55.6 ± 1.8 |
| H3K9me3 | 48.2 ± 1.8 | 46.7 ± 4.4 | 44.7 ± 1.8 | 45.3 ± 1.6 | 37.5 ± 2.6 | 43.5 ± 1.9 | 45.5 ± 1.9 | 44.1 ± 1.7 | 50.9 ± 1.3 | 48.0 ± 3.7 |
| H4K20me1 | 66.9 ± 0.6 | 65.4 ± 1.1 | 65.0 ± 1.4 | 60.6 ± 1.6 | 58.0 ± 0.9 | 57.2 ± 1.2 | 59.0 ± 2.0 | 63.4 ± 0.6 | 67.0 ± 0.6 | 65.2 ± 1.0 |
| Enhancer | 54.5 ± 0.6 | 57.5 ± 2.3 | 52.7 ± 1.2 | 61.4 ± 1.0 | 47.5 ± 0.6 | 48.0 ± 0.8 | 49.0 ± 0.9 | 51.9 ± 0.9 | 59.4 ± 1.3 | 55.3 ± 2.0 |
| Enhancer type | 51.2 ± 1.3 | 54.1 ± 1.3 | 48.4 ± 1.2 | 57.3 ± 1.3 | 44.1 ± 1.0 | 46.1 ± 0.9 | 45.9 ± 1.1 | 48.1 ± 0.9 | 54.7 ± 1.7 | 51.0 ± 2.2 |
| Promoter all | 77.4 ± 0.6 | 78.0 ± 1.2 | 76.1 ± 0.9 | 74.5 ± 1.2 | 69.3 ± 1.6 | 70.7 ± 1.7 | 72.2 ± 1.4 | 72.1 ± 1.1 | 79.5 ± 0.5 | 76.5 ± 0.9 |
| Promoter non-TATA | 77.9 ± 0.9 | 78.5 ± 0.9 | 77.3 ± 1.0 | 76.3 ± 1.2 | 72.3 ± 1.3 | 74.0 ± 1.2 | 74.6 ± 0.9 | 73.9 ± 1.8 | 80.1 ± 0.5 | 78.6 ± 0.7 |
| Promoter TATA | 90.6 ± 2.0 | 91.9 ± 2.8 | 94.4 ± 1.6 | 79.3 ± 2.6 | 64.8 ± 4.4 | 86.8 ± 2.3 | 85.3 ± 3.4 | 89.1 ± 4.1 | 95.0 ± 0.9 | 86.2 ± 2.4 |
| Splice acceptor | 85.1 ± 0.3 | 96.5 ± 0.4 | 95.8 ± 0.3 | 74.9 ± 0.7 | 81.5 ± 4.9 | 90.6 ± 1.5 | 93.9 ± 1.2 | 81.2 ± 1.2 | 96.4 ± 0.3 | 95.1 ± 0.6 |
| Splice site all | 87.6 ± 0.3 | 96.8 ± 0.3 | 96.4 ± 0.3 | 73.9 ± 1.1 | 85.4 ± 5.3 | 94.1 ± 0.6 | 94.2 ± 1.2 | 84.9 ± 1.5 | 96.6 ± 0.3 | 95.9 ± 0.3 |
| Splice donor | 86.8 ± 0.6 | 97.6 ± 0.3 | 97.0 ± 0.2 | 78.0 ± 0.7 | 94.3 ± 2.4 | 94.4 ± 2.6 | 96.4 ± 1.0 | 84.2 ± 0.9 | 97.7 ± 0.2 | 97.1 ± 0.2 |
| **Average rank (all)** | **4.22** | **4.33** | **7.28** | **7.28** | **13.06** | **11.94** | **10.28** | **9.39** | **1.44** | **5.72** |

# 4 CONCLUSION

Standard Transformer encoders continue to improve through practical architectural and training modernizations that increase computational efficiency, robustness, and long-context capability. In this work, we study how these advances transfer to genomic sequence modeling by adapting ModernBERT for DNA pretraining and introducing ModernGENA.

We evaluate ModernGENA along two axes that matter in practice: efficiency and downstream transfer. Under controlled settings, we benchmark inference throughput across a range of sequence lengths and evaluate downstream performance on the Nucleotide Transformer benchmark. Modern-GENA supports FlashAttention-based implementations and achieves higher inference throughput in our experiments, while delivering strong benchmark performance. ModernGENA ranks first among encoder-only models with comparable size and second overall in our evaluation suite, reflecting a favorable balance between efficiency and quality. Finally, because ModernGENA builds on the ModernBERT design, it is also intended to support long genomic contexts, a key requirement in many genomics applications.

Our results also point to a broader and still unresolved evaluation question in this area. Although the Nucleotide Transformer benchmark is one of the most widely used suites for comparing DNA foundation models, the absolute differences between architectures, and even between substantially different model scales, often appear modest. In practice, part of this gap can sometimes be reduced through careful tuning of fine-tuning hyperparameters and training protocols, which makes it difficult to draw strong conclusions from small metric differences alone. This raises an important open question: to what extent does increasing model scale translate into reliable downstream improvements for genomic sequence modeling, beyond what can be achieved with smaller models and strong training recipes? Answering this likely requires deeper analysis of benchmark characteristics and fine-tuning variance, and it motivates broader evaluation, either by incorporating additional benchmarks that stress different regimes (including longer-range context) or by reconsidering when scaling is actually necessary for genomic sequence understanding. In this context, we position ModernGENA as both a strong, reproducible baseline and a practical component for building and testing more complex architectures: it is efficient in throughput, adapted to long sequences, and intended to make future architectural experiments more systematic and comparable.

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

## APPENDIX A. ARCHITECTURE CONFIGURATION

Additional architecture configuration details for ModernGENA Base and Large are provided in Table 3.

Table 3: Architecture configuration.

| Parameter | ModernGENA Base | ModernGENA Large |
|---|---|---|
| parameters | 135M | 377M |
| num_hidden_layers | 22 | 28 |
| hidden_size | 768 | 1024 |
| intermediate_size | 1152 | 2624 |
| num_attention_heads | 12 | 16 |
| attn_out_dropout_prob | 0.1 | 0.1 |
| mlp_layer | glu | glu |
| mlp_out_bias | false | false |
| activation_function | gelu | gelu |
| rotary_emb_base | 10000 | 10000 |
| sliding_window | 128 | 128 |
| global_attn_every_n_layers | 3 | 3 |

## APPENDIX B. INFERENCE EFFICIENCY DETAILS

Inference efficiency is evaluated on an NVIDIA A100 (80 GB) using fixed-length sequences, reporting throughput (tokens/s) for models in the *primary baseline set*. The resulting batch sizes and tokens/s are reported in Table 4.

Table 4: **Inference efficiency evaluation.** For each sequence length $L$, the table reports the maximum batch size that fits in memory (bs) and throughput in tokens/s (mean over 10 runs).

| Model | \multicolumn{12}{c}{Sequence length $L$ (tokens)} | | | | | | | | | | | |
|---|---|---|---|---|---|---|---|---|---|---|---|---|
| | \multicolumn{2}{c}{256} | | \multicolumn{2}{c}{512} | | \multicolumn{2}{c}{1024} | | \multicolumn{2}{c}{2048} | | \multicolumn{2}{c}{4096} | | \multicolumn{2}{c}{9192} | |
| | bs | tokens/s | bs | tokens/s | bs | tokens/s | bs | tokens/s | bs | tokens/s | bs | tokens/s |
| **ModernGENA (FA2 + autocast bf16) (135M)** | 3631 | **552.44** | 1811 | **529.18** | 901 | **507.61** | 446 | **479.04** | 218 | **425.88** | 92 | **332.54** |
| ModernGENA (135M) | 3631 | 78.12 | 1811 | 77.73 | 901 | 77.20 | 446 | 76.38 | 218 | 74.83 | 92 | 71.15 |
| GENA (110M) | 6637 | 91.78 | 2206 | 81.98 | – | – | – | – | – | – | – | – |
| DNABERT-2 (117M) | 3802 | 68.57 | 1433 | 62.63 | 376 | 53.42 | 90 | 40.68 | 15 | 27.57 | 5 | 15.48 |
| NTv2 (100M) | 5941 | 70.88 | 1845 | 60.28 | 519 | 46.46 | 132 | 31.72 | 26 | 19.02 | 5 | 9.37 |
| **ModernGENA Large (FA2 + autocast bf16) (377M)** | 2721 | **223.77** | 1356 | **218.23** | 673 | **212.24** | 332 | **202.51** | 161 | **185.82** | 67 | **153.77** |
| ModernGENA Large (377M) | 2721 | 26.20 | 1356 | 26.11 | 673 | 26.01 | 332 | 25.85 | 161 | 25.56 | 67 | 24.86 |
| GENA Large (336M) | 4922 | 27.13 | 1633 | 24.79 | – | – | – | – | – | – | – | – |
| NTv2 (250M) | 4107 | 33.47 | 1649 | 30.62 | 486 | 26.07 | 127 | 20.05 | 26 | 13.49 | 5 | 7.35 |

## APPENDIX C. HYPERPARAMETERS PER NT TASK

ModernGENA is fine-tuned using a grid search over learning rates $\{1 \times 10^{-5}, 3 \times 10^{-5}, 5 \times 10^{-5}\}$, weight decay $\{1 \times 10^{-4}, 1 \times 10^{-3}, 1 \times 10^{-2}\}$, and effective batch sizes $\{32, 64\}$. GENA is fine-tuned using a grid search over learning rates $\{1 \times 10^{-5}, 3 \times 10^{-5}, 5 \times 10^{-5}\}$, weight decay $\{1 \times 10^{-4}, 1 \times 10^{-3}\}$, and effective batch sizes $\{32, 64\}$. The selected hyperparameters for each task are reported in Table 5.

Table 5: **Selected hyperparameters per NT task.** lr: learning rate; wd: weight decay; bs: batch size.

| Task | \multicolumn{3}{c}{GENA} | | | \multicolumn{3}{c}{ModernGENA} | | |
|---|---|---|---|---|---|---|
| | lr | wd | bs | lr | wd | bs |
| H2AFZ | $3 \times 10^{-5}$ | $1 \times 10^{-3}$ | 32 | $5 \times 10^{-5}$ | $1 \times 10^{-4}$ | 64 |
| H3K27ac | $3 \times 10^{-5}$ | $1 \times 10^{-4}$ | 64 | $5 \times 10^{-5}$ | $1 \times 10^{-3}$ | 32 |
| H3K27me3 | $5 \times 10^{-5}$ | $1 \times 10^{-3}$ | 32 | $5 \times 10^{-5}$ | $1 \times 10^{-3}$ | 32 |
| H3K36me3 | $3 \times 10^{-5}$ | $1 \times 10^{-4}$ | 64 | $5 \times 10^{-5}$ | $1 \times 10^{-3}$ | 32 |
| H3K4me1 | $3 \times 10^{-5}$ | $1 \times 10^{-4}$ | 32 | $1 \times 10^{-5}$ | $1 \times 10^{-4}$ | 32 |
| H3K4me2 | $3 \times 10^{-5}$ | $1 \times 10^{-4}$ | 32 | $1 \times 10^{-5}$ | $1 \times 10^{-2}$ | 64 |
| H3K4me3 | $3 \times 10^{-5}$ | $1 \times 10^{-3}$ | 32 | $5 \times 10^{-5}$ | $1 \times 10^{-2}$ | 32 |
| H3K9ac | $3 \times 10^{-5}$ | $1 \times 10^{-3}$ | 64 | $1 \times 10^{-5}$ | $1 \times 10^{-4}$ | 32 |
| H3K9me3 | $3 \times 10^{-5}$ | $1 \times 10^{-3}$ | 64 | $3 \times 10^{-5}$ | $1 \times 10^{-4}$ | 64 |
| H4K20me1 | $5 \times 10^{-5}$ | $1 \times 10^{-3}$ | 32 | $3 \times 10^{-5}$ | $1 \times 10^{-2}$ | 32 |
| Enhancer | $3 \times 10^{-5}$ | $1 \times 10^{-4}$ | 32 | $5 \times 10^{-5}$ | $1 \times 10^{-3}$ | 32 |
| Enhancer type | $3 \times 10^{-5}$ | $1 \times 10^{-3}$ | 64 | $3 \times 10^{-5}$ | $1 \times 10^{-3}$ | 32 |
| Promoter all | $5 \times 10^{-5}$ | $1 \times 10^{-3}$ | 64 | $1 \times 10^{-5}$ | $1 \times 10^{-4}$ | 32 |
| Promoter non-TATA | $5 \times 10^{-5}$ | $1 \times 10^{-4}$ | 32 | $3 \times 10^{-5}$ | $1 \times 10^{-4}$ | 64 |
| Promoter TATA | $5 \times 10^{-5}$ | $1 \times 10^{-3}$ | 32 | $5 \times 10^{-5}$ | $1 \times 10^{-3}$ | 32 |
| Splice acceptor | $5 \times 10^{-5}$ | $1 \times 10^{-4}$ | 32 | $5 \times 10^{-5}$ | $1 \times 10^{-4}$ | 64 |
| Splice site all | $5 \times 10^{-5}$ | $1 \times 10^{-3}$ | 32 | $5 \times 10^{-5}$ | $1 \times 10^{-4}$ | 64 |
| Splice donor | $5 \times 10^{-5}$ | $1 \times 10^{-3}$ | 64 | $5 \times 10^{-5}$ | $1 \times 10^{-3}$ | 64 |

## APPENDIX D. GENOMIC ASSEMBLIES FOR MULTISPECIES PRETRAINING

Table 6: **List of genomic assemblies** used to create the multispecies pretraining dataset. Assembly accessions correspond to the genome and annotation identifiers processed by the NCBI Eukaryotic Genome Annotation Pipeline.

| Assembly | Species |
|---|---|
| GCF_021347895.1 | Acanthochromis polyacanthus |
| GCF_904848185.1 | Acanthopagrus latus |

| Assembly | Species |
|---|---|
| GCF_027475565.1 | Acinonyx jubatus |
| GCF_902713425.1 | Acipenser ruthenus |
| GCF_903995435.1 | Acomys russatus |
| GCF_020745825.1 | Agelaius phoeniceus |
| GCF_028640845.1 | Ahaetulla prasina |
| GCF_002007445.2 | Ailuropoda melanoleuca |
| GCF_030867095.1 | Alligator mississippiensis |
| GCF_017589495.1 | Alosa alosa |
| GCF_018492685.1 | Alosa sapidissima |
| GCF_010909765.2 | Amblyraja radiata |
| GCF_036373705.1 | Amia ocellicauda |
| GCF_027887145.1 | Ammospiza caudacuta |
| GCF_027579445.1 | Ammospiza nelsoni |
| GCF_022539595.1 | Amphiprion ocellaris |
| GCF_900324465.2 | Anabas testudineus |
| GCF_963932015.1 | Anas acuta |
| GCF_015476345.1 | Anas platyrhynchos |
| GCF_013347855.1 | Anguilla anguilla |
| GCF_018555375.3 | Anguilla rostrata |
| GCF_035594765.1 | Anolis carolinensis |
| GCF_037176765.1 | Anolis sagrei |
| GCF_031753505.1 | Anomalospiza imberbis |
| GCF_027596085.1 | Anoplopoma fimbria |
| GCF_040182565.1 | Anser cygnoides |
| GCF_016432865.1 | Antechinus flavipes |
| GCF_040054535.1 | Antennarius striatus |
| GCF_041296385.1 | Aphelocoma coerulescens |
| GCF_947179515.1 | Apodemus sylvaticus |
| GCF_036417845.1 | Apteryx mantelli |
| GCF_020740795.1 | Apus apus |
| GCF_900496995.4 | Aquila chrysaetos chrysaetos |
| GCF_007364275.1 | Archocentrus centrarchus |
| GCF_011762505.1 | Arvicanthis niloticus |
| GCF_903992535.2 | Arvicola amphibius |
| GCF_900246225.1 | Astatotilapia calliptera |
| GCF_929443795.1 | Astur gentilis |
| GCF_023375975.1 | Astyanax mexicanus |
| GCF_009819795.1 | Aythya fuligula |
| GCF_949987535.1 | Balaenoptera acutorostrata |
| GCF_009873245.2 | Balaenoptera musculus |
| GCF_028023285.1 | Balaenoptera ricei |
| GCF_900634795.4 | Betta splendens |
| GCF_027579735.1 | Bombina bombina |
| GCF_000247795.1 | Bos indicus |
| GCF_003369695.1 | Bos indicus x Bos taurus |
| GCF_032452875.1 | Bos javanicus |
| GCF_002263795.3 | Bos taurus |
| GCF_040956055.1 | Brachionichthys hirsutus |
| GCF_023856365.1 | Brienomyrus brachyistius |
| GCF_019923935.1 | Bubalus bubalis |
| GCF_029407905.1 | Bubalus kerabau |
| GCF_023091745.1 | Budorcas taxicolor |
| GCF_905171765.1 | Bufo bufo |
| GCF_014858855.1 | Bufo gargarizans |
| GCF_011100555.1 | Callithrix jacchus |
| GCF_036013445.1 | Caloenas nicobarica |
| GCF_003957555.1 | Calypte anna |
| GCF_901933205.1 | Camarhynchus parvulus |

| Assembly | Species |
| --- | --- |
| GCF_036321535.1 | Camelus dromedarius |
| GCF_009834535.1 | Camelus ferus |
| GCF_035149785.1 | Candoia aspera |
| GCF_003254725.2 | Canis lupus dingo |
| GCF_011100685.1 | Canis lupus familiaris |
| GCF_001704415.2 | Capra hircus |
| GCF_032405125.1 | Capricornis sumatraensis |
| GCF_003368295.1 | Carassius auratus |
| GCF_963082965.1 | Carassius carassius |
| GCF_023724105.1 | Carassius gibelio |
| GCF_017639515.1 | Carcharodon carcharias |
| GCF_023653815.1 | Caretta caretta |
| GCF_009819885.2 | Catharus ustulatus |
| GCF_030273125.1 | Centropristis striata |
| GCF_019320065.1 | Cervus canadensis |
| GCF_910594005.1 | Cervus elaphus |
| GCF_033026475.1 | Channa argus |
| GCF_024489055.1 | Chanodichthys erythropterus |
| GCF_902362185.1 | Chanos chanos |
| GCF_018320785.1 | Cheilinus undulatus |
| GCF_017976325.1 | Chelmon rostratus |
| GCF_015237465.2 | Chelonia mydas |
| GCF_004010195.1 | Chiloscyllium plagiosum |
| GCF_950005125.1 | Chionomys nivalis |
| GCF_009829145.1 | Chiroxiphia lanceolata |
| GCF_015220235.1 | Choloepus didactylus |
| GCF_963924245.1 | Chroicocephalus ridibundus |
| GCF_011386835.1 | Chrysemys picta bellii |
| GCF_963662255.1 | Cinclus cinclus |
| GCF_024256425.1 | Clarias gariepinus |
| GCF_900700415.2 | Clupea harengus |
| GCF_028858725.1 | Colius striatus |
| GCF_033807715.1 | Cololabis saira |
| GCF_036013475.1 | Columba livia |
| GCF_963514075.1 | Conger conger |
| GCF_020615455.1 | Coregonus clupeaformis |
| GCF_000738735.6 | Corvus cornix cornix |
| GCF_020740725.1 | Corvus hawaiiensis |
| GCF_009650955.1 | Corvus moneduloides |
| GCF_030265065.1 | Corythoichthys intestinalis |
| GCF_900634415.1 | Cottoperca gobio |
| GCF_001577835.2 | Coturnix japonica |
| GCF_003668045.3 | Cricetulus griseus |
| GCF_019924925.1 | Ctenopharyngodon idella |
| GCF_017976375.1 | Cuculus canorus |
| GCF_009769545.1 | Cyclopterus lumpus |
| GCF_013377495.2 | Cygnus atratus |
| GCF_009769625.2 | Cygnus olor |
| GCF_027409185.1 | Cynocephalus volans |
| GCF_000523025.1 | Cynoglossus semilaevis |
| GCF_018340385.1 | Cyprinus carpio |
| GCF_033118175.1 | Dama dama |
| GCF_903798145.1 | Danio aesculapii |
| GCF_000002035.6 | Danio rerio |
| GCF_030445035.1 | Dasypus novemcinctus |
| GCF_949987515.1 | Delphinus delphis |
| GCF_900700375.1 | Denticeps clupeoides |
| GCF_009764565.3 | Dermochelys coriacea |

| Assembly | Species |
|---|---|
| GCF_022682495.1 | Desmodus rotundus |
| GCF_020826845.1 | Diceros bicornis minor |
| GCF_030265055.1 | Doryrhamphus excisus |
| GCF_036370855.1 | Dromaius novaehollandiae |
| GCF_019393635.1 | Dromiciops gliroides |
| GCF_014839835.1 | Dryobates pubescens |
| GCF_027744805.1 | Dunckerocampus dactyliophorus |
| GCF_900963305.1 | Echeneis naucrates |
| GCF_013358815.1 | Electrophorus electricus |
| GCF_036324505.1 | Eleginops maclovinus |
| GCF_024166365.1 | Elephas maximus indicus |
| GCF_035609145.1 | Eleutherodactylus coqui |
| GCF_023053635.1 | Elgaria multicarinata webbii |
| GCF_028017835.1 | Emys orbicularis |
| GCF_034702125.1 | Engraulis encrasicolus |
| GCF_033978785.1 | Entelurus aequoreus |
| GCF_011397635.1 | Epinephelus fuscoguttatus |
| GCF_005281545.1 | Epinephelus lanceolatus |
| GCF_006386435.1 | Epinephelus moara |
| GCF_027574615.1 | Eptesicus fuscus |
| GCF_016077325.2 | Equus asinus |
| GCF_002863925.1 | Equus caballus |
| GCF_021613505.1 | Equus quagga |
| GCF_950295315.1 | Erinaceus europaeus |
| GCF_900747795.2 | Erpetoichthys calabaricus |
| GCF_028021215.1 | Eschrichtius robustus |
| GCF_011004845.1 | Esox lucius |
| GCF_013103735.1 | Etheostoma cragini |
| GCF_008692095.1 | Etheostoma spectabile |
| GCF_028564815.1 | Eubalaena glacialis |
| GCF_028583425.1 | Eublepharis macularius |
| GCF_029931775.1 | Euleptes europaea |
| GCF_023638135.1 | Falco biarmicus |
| GCF_023634085.1 | Falco cherrug |
| GCF_017639655.2 | Falco naumanni |
| GCF_023634155.1 | Falco peregrinus |
| GCF_015220075.1 | Falco rusticolus |
| GCF_018350175.1 | Felis catus |
| GCF_000247815.1 | Ficedula albicollis |
| GCF_011125445.2 | Fundulus heteroclitus |
| GCF_026213295.1 | Gadus chalcogrammus |
| GCF_031168955.1 | Gadus macrocephalus |
| GCF_902167405.1 | Gadus morhua |
| GCF_016699485.2 | Gallus gallus |
| GCF_019740435.1 | Gambusia affinis |
| GCF_016920845.1 | Gasterosteus aculeatus aculeatus |
| GCF_030936135.1 | Gavia stellata |
| GCF_902459505.1 | Geotrypetes seraphini |
| GCF_021462225.1 | Girardinichthys multiradiatus |
| GCF_963455315.1 | Globicephala melas |
| GCF_007399415.2 | Gopherus evgoodei |
| GCF_025201925.1 | Gopherus flavomarginatus |
| GCF_029281585.2 | Gorilla gorilla gorilla |
| GCF_900634775.1 | Gouania willdenowi |
| GCF_016433145.1 | Gracilinanus agilis |
| GCF_028858705.1 | Grus americana |
| GCF_018139145.2 | Gymnogyps californianus |
| GCF_027477595.1 | Haemorhous mexicanus |

| Assembly | Species |
|---|---|
| GCF_026419915.1 | Harpia harpyja |
| GCF_019097595.1 | Hemibagrus wyckioides |
| GCF_027244095.1 | Hemicordylus capensis |
| GCF_020745735.1 | Hemiscyllium ocellatum |
| GCF_035084215.1 | Heptranchias perlo |
| GCF_036365525.1 | Heterodontus francisci |
| GCF_032191835.1 | Heteronotia binoei |
| GCF_025434085.1 | Hippocampus zosterae |
| GCF_009819705.1 | Hippoglossus hippoglossus |
| GCF_022539355.2 | Hippoglossus stenolepis |
| GCF_030028045.1 | Hippopotamus amphibius kiboko |
| GCF_015227805.2 | Hirundo rustica |
| GCF_000001405.40 | Homo sapiens |
| GCF_029633855.1 | Hoplias malabaricus |
| GCF_029499605.1 | Hyla sarda |
| GCF_030144855.1 | Hypanus sabinus |
| GCF_040937935.1 | Hyperolius riggenbachi |
| GCF_021917145.1 | Hypomesus transpacificus |
| GCF_023375685.1 | Ictalurus furcatus |
| GCF_001660625.3 | Ictalurus punctatus |
| GCF_016881025.1 | Ictidomys tridecemlineatus |
| GCF_027791375.1 | Indicator indicator |
| GCF_020740685.1 | Jaculus jaculus |
| GCF_026419965.1 | Kogia breviceps |
| GCF_001649575.2 | Kryptolebias marmoratus |
| GCF_022985175.1 | Labeo rohita |
| GCF_963930695.1 | Labrus bergylta |
| GCF_963584025.1 | Labrus mixtus |
| GCF_009819535.1 | Lacerta agilis |
| GCF_949774975.1 | Lagenorhynchus albirostris |
| GCF_023343835.1 | Lagopus muta |
| GCF_029633865.1 | Lampris incognitus |
| GCF_000972845.2 | Larimichthys crocea |
| GCF_001640805.2 | Lates calcarifer |
| GCF_037157495.1 | Lathamus discolor |
| GCF_037176945.1 | Latimeria chalumnae |
| GCF_020740605.2 | Lemur catta |
| GCF_018350155.1 | Leopardus geoffroyi |
| GCF_033115175.1 | Lepus europaeus |
| GCF_015708825.1 | Lethenteron reissneri |
| GCF_028641065.1 | Leucoraja erinaceus |
| GCF_963576545.1 | Limanda limanda |
| GCF_005870125.1 | Lonchura striata |
| GCF_030014295.1 | Loxodonta africana |
| GCF_902655055.1 | Lutra lutra |
| GCF_007474595.2 | Lynx canadensis |
| GCF_037993035.1 | Macaca fascicularis |
| GCF_003339765.1 | Macaca mulatta |
| GCF_024542745.1 | Macaca thibetana thibetana |
| GCF_027887155.1 | Malaclemys terrapin pileata |
| GCF_030020395.1 | Manis pentadactyla |
| GCF_900324485.2 | Mastacembelus armatus |
| GCF_008632895.1 | Mastomys coucha |
| GCF_020497125.1 | Mauremys mutica |
| GCF_016161935.1 | Mauremys reevesii |
| GCF_000238955.4 | Maylandia zebra |
| GCF_018812025.1 | Megalobrama amblycephala |
| GCF_013368585.1 | Megalops cyprinoides |

| Assembly | Species |
|---|---|
| GCF_017639745.1 | Melanotaenia boesemani |
| GCF_000146605.3 | Meleagris gallopavo |
| GCF_922984935.1 | Meles meles |
| GCF_012275295.1 | Melopsittacus undulatus |
| GCF_028018845.1 | Melospiza georgiana |
| GCF_035770615.1 | Melospiza melodia melodia |
| GCF_030254825.1 | Meriones unguiculatus |
| GCF_025265405.1 | Mesoplodon densirostris |
| GCF_901765095.1 | Microcaecilia unicolor |
| GCF_000165445.2 | Microcebus murinus |
| GCF_021292245.1 | Micropterus dolomieu |
| GCF_000317375.1 | Microtus ochrogaster |
| GCF_027580225.1 | Misgurnus anguillicaudatus |
| GCF_963921235.1 | Mobula hypostoma |
| GCF_037042795.1 | Molothrus aeneus |
| GCF_012460135.2 | Molothrus ater |
| GCF_027887165.1 | Monodelphis domestica |
| GCF_015832195.1 | Motacilla alba alba |
| GCF_022458985.1 | Mugil cephalus |
| GCF_963930625.1 | Muntiacus reevesi |
| GCF_900094665.2 | Mus caroli |
| GCF_000001635.27 | Mus musculus |
| GCF_900095145.1 | Mus pahari |
| GCF_009829155.1 | Mustela erminea |
| GCF_030435805.1 | Mustela lutreola |
| GCF_022355385.1 | Mustela nigripes |
| GCF_963259705.1 | Myotis daubentonii |
| GCF_902150065.1 | Myripristis murdjan |
| GCF_040869285.1 | Myxine glutinosa |
| GCF_019703515.2 | Myxocyprinus asiaticus |
| GCF_014905685.2 | Nematolebias whitei |
| GCF_027579695.1 | Neoarius graeffei |
| GCF_028018385.1 | Neofelis nebulosa |
| GCF_020171115.1 | Neogale vison |
| GCF_002201575.2 | Neomonachus schauinslandi |
| GCF_033978795.1 | Nerophis ophidion |
| GCF_006542625.1 | Nomascus leucogenys |
| GCF_009762535.1 | Notolabrus celidotus |
| GCF_002078875.1 | Numida meleagris |
| GCF_013368605.1 | Nyctibius grandis |
| GCF_027406575.1 | Nycticebus coucang |
| GCF_030435755.1 | Ochotona princeps |
| GCF_029582105.1 | Oenanthe melanoleuca |
| GCF_021184085.1 | Oncorhynchus gorbuscha |
| GCF_023373465.1 | Oncorhynchus keta |
| GCF_002021735.2 | Oncorhynchus kisutch |
| GCF_036934945.1 | Oncorhynchus masou masou |
| GCF_013265735.2 | Oncorhynchus mykiss |
| GCF_034236695.1 | Oncorhynchus nerka |
| GCF_018296145.1 | Oncorhynchus tshawytscha |
| GCF_903995425.1 | Onychomys torridus |
| GCF_012432095.1 | Onychostoma macrolepis |
| GCF_937001465.1 | Orcinus orca |
| GCF_013358895.1 | Oreochromis aureus |
| GCF_001858045.2 | Oreochromis niloticus |
| GCF_004115215.2 | Ornithorhynchus anatinus |
| GCF_009806435.1 | Oryctolagus cuniculus |
| GCF_002234675.1 | Oryzias latipes |

| Assembly | Species |
| --- | --- |
| GCF_002922805.2 | Oryzias melastigma |
| GCF_963692335.1 | Osmerus eperlanus |
| GCF_038355195.1 | Osmerus mordax |
| GCF_016772045.2 | Ovis aries |
| GCF_011077185.1 | Oxyura jamaicensis |
| GCF_029289425.2 | Pan paniscus |
| GCF_028858775.2 | Pan troglodytes |
| GCF_027358585.1 | Pangasianodon hypophthalmus |
| GCF_018350215.1 | Panthera leo |
| GCF_028533385.1 | Panthera onca |
| GCF_018350195.1 | Panthera tigris |
| GCF_023721935.1 | Panthera uncia |
| GCF_008728515.1 | Papio anubis |
| GCF_900634625.1 | Parambassis ranga |
| GCF_001522545.3 | Parus major |
| GCF_036417665.1 | Passer domesticus |
| GCF_036971685.1 | Patagioenas fasciata |
| GCF_036321145.2 | Pelmatolapia mariae |
| GCF_036172605.1 | Pelobates fuscus |
| GCF_004354835.1 | Perca flavescens |
| GCF_010015445.1 | Perca fluviatilis |
| GCF_009829125.3 | Periophthalmus magnuspinnatus |
| GCF_023159225.1 | Perognathus longimembris pacificus |
| GCF_949786415.1 | Peromyscus eremicus |
| GCF_004664715.2 | Peromyscus leucopus |
| GCF_003704035.1 | Peromyscus maniculatus bairdii |
| GCF_010993605.1 | Petromyzon marinus |
| GCF_016906955.1 | Phacochoerus africanus |
| GCF_963921805.1 | Phalacrocorax carbo |
| GCF_963924675.1 | Phocoena phocoena |
| GCF_008692025.1 | Phocoena sinus |
| GCF_024500275.1 | Phycodurus eques |
| GCF_024500385.1 | Phyllopteryx taeniolatus |
| GCF_004126475.2 | Phyllostomus discolor |
| GCF_002837175.3 | Physeter macrocephalus |
| GCF_002776525.5 | Piliocolobus tephrosceles |
| GCF_949316205.1 | Platichthys flesus |
| GCF_008729295.1 | Plectropomus leopardus |
| GCF_031143425.1 | Pleurodeles waltl |
| GCF_947347685.1 | Pleuronectes platessa |
| GCF_004329235.1 | Podarcis muralis |
| GCF_027172205.1 | Podarcis raffonei |
| GCF_030490865.1 | Poecile atricapillus |
| GCF_000633615.1 | Poecilia reticulata |
| GCF_015220805.1 | Pogoniulus pusillus |
| GCF_017654505.1 | Polyodon spathula |
| GCF_016835505.1 | Polypterus senegalus |
| GCF_028885655.2 | Pongo abelii |
| GCF_028885625.2 | Pongo pygmaeus |
| GCF_021018805.1 | Prinia subflava |
| GCF_016509475.1 | Prionailurus bengalensis |
| GCF_022837055.1 | Prionailurus viverrinus |
| GCF_009764475.1 | Pristis pectinata |
| GCF_019279795.1 | Protopterus annectens |
| GCF_902827115.1 | Pseudochaenichthys georgianus |
| GCF_029220125.1 | Pseudoliparis swirei |
| GCF_028390025.1 | Pseudophryne corroboree |
| GCF_036250125.1 | Pseudopipra pipra |

| Assembly | Species |
| --- | --- |
| GCF_024679245.1 | Pseudorasbora parva |
| GCF_039906515.1 | Pseudorca crassidens |
| GCF_949316345.1 | Pungitius pungitius |
| GCF_018831695.1 | Puntigrus tetrazona |
| GCF_015220715.1 | Pygocentrus nattereri |
| GCF_905171775.1 | Rana temporaria |
| GCF_036323735.1 | Rattus norvegicus |
| GCF_011064425.1 | Rattus rattus |
| GCF_028389875.1 | Rhea pennata |
| GCF_901001135.1 | Rhinatrema bivittatum |
| GCF_021869965.1 | Rhincodon typus |
| GCF_030035675.1 | Rhineura floridana |
| GCF_004115265.2 | Rhinolophus ferrumequinum |
| GCF_007565055.1 | Rhinopithecus roxellana |
| GCF_028500815.1 | Rissa tridactyla |
| GCF_036850765.1 | Saccopteryx bilineata |
| GCF_036850995.1 | Saccopteryx leptura |
| GCF_902148845.1 | Salarias fasciatus |
| GCF_905237065.1 | Salmo salar |
| GCF_901001165.1 | Salmo trutta |
| GCF_029448725.1 | Salvelinus fontinalis |
| GCF_016432855.1 | Salvelinus namaycush |
| GCF_002910315.2 | Salvelinus sp. IW2-2015 |
| GCF_008315115.2 | Sander lucioperca |
| GCF_902635505.1 | Sarcophilus harrisii |
| GCF_963854185.1 | Sardina pilchardus |
| GCF_020382885.2 | Scatophagus argus |
| GCF_019175285.1 | Sceloporus undulatus |
| GCF_902686445.1 | Sciurus carolinensis |
| GCF_900964775.1 | Scleropages formosus |
| GCF_027409825.1 | Scomber japonicus |
| GCF_963691925.1 | Scomber scombrus |
| GCF_022379125.1 | Scophthalmus maximus |
| GCF_902713615.1 | Scyliorhinus canicula |
| GCF_015220745.1 | Sebastes umbrosus |
| GCF_022539315.1 | Serinus canaria |
| GCF_021018895.1 | Seriola aureovittata |
| GCF_014805685.1 | Silurus meridionalis |
| GCF_020085105.1 | Siniperca chuatsi |
| GCF_019176455.1 | Solea senegalensis |
| GCF_958295425.1 | Solea solea |
| GCF_027595985.1 | Sorex araneus |
| GCF_900880675.1 | Sparus aurata |
| GCF_027358695.1 | Spea bombifrons |
| GCF_902148855.1 | Sphaeramia orbicularis |
| GCF_021028975.2 | Sphaerodactylus townsendi |
| GCF_030684315.1 | Stegostoma tigrinum |
| GCF_004027225.2 | Strigops habroptila |
| GCF_040807025.1 | Struthio camelus |
| GCF_024139225.1 | Suncus etruscus |
| GCF_006229205.1 | Suricata suricatta |
| GCF_000003025.6 | Sus scrofa |
| GCF_009819655.1 | Sylvia atricapilla |
| GCF_028878055.3 | Symphalangus syndactylus |
| GCF_027744825.2 | Synchiropus splendidus |
| GCF_019802595.1 | Syngnathoides biaculeatus |
| GCF_901709675.1 | Syngnathus acus |
| GCF_024217435.2 | Syngnathus scovelli |

| Assembly | Species |
|---|---|
| GCF_033458585.1 | Syngnathus typhle |
| GCF_015852505.1 | Tachyglossus aculeatus |
| GCF_022655615.1 | Tachysurus fulvidraco |
| GCF_030014155.1 | Tachysurus vachellii |
| GCF_003957565.2 | Taeniopygia guttata |
| GCF_003711565.1 | Takifugu flavidus |
| GCF_901000725.2 | Takifugu rubripes |
| GCF_902500255.1 | Thalassophryne amazonica |
| GCF_009769535.1 | Thamnophis elegans |
| GCF_003255815.1 | Theropithecus gelada |
| GCF_914725855.1 | Thunnus albacares |
| GCF_910596095.1 | Thunnus maccoyii |
| GCF_963924715.1 | Thunnus thynnus |
| GCF_035046505.1 | Tiliqua scincoides |
| GCF_017976425.1 | Toxotes jaculatrix |
| GCF_013100865.1 | Trachemys scripta elegans |
| GCF_030014385.1 | Trichomycterus rosablanca |
| GCF_011100635.1 | Trichosurus vulpecula |
| GCF_015846415.1 | Triplophysa dalaica |
| GCF_024868665.1 | Triplophysa rosa |
| GCF_011762595.1 | Tursiops truncatus |
| GCF_023065955.2 | Ursus arctos |
| GCF_026979565.1 | Vidua chalybeata |
| GCF_024509145.1 | Vidua macroura |
| GCF_018345385.1 | Vulpes lagopus |
| GCF_017654675.1 | Xenopus laevis |
| GCF_000004195.4 | Xenopus tropicalis |
| GCF_016859285.1 | Xiphias gladius |
| GCF_001444195.1 | Xiphophorus couchianus |
| GCF_003331165.1 | Xiphophorus hellerii |
| GCF_002775205.1 | Xiphophorus maculatus |
| GCF_025860055.1 | Xyrauchen texanus |
| GCF_009762305.2 | Zalophus californianus |
| GCF_028769735.1 | Zonotrichia leucophrys gambelii |
| GCF_963506605.1 | Zootoca vivipara |

## APPENDIX E. DECLARATION OF LLM USAGE.

Large Language Models (LLMs) were used solely to improve the readability and clarity of the manuscript text. No parts of the analysis, results, or conclusions were generated by LLMs.

