# OpenReview forum: "Back to BERT in 2026: ModernGENA as a Strong, Efficient Baseline for DNA Foundation Models"
_ICLR.cc/2026/Workshop/FM4Science — ICLR 2026 Workshop FM4Science Poster_

### Official Review · Reviewer_7dZJ · 2026-02-15
**Strong, Efficient DNA Baseline via Modernized BERT - Clear Accept**

**Rating:** 8
**Confidence:** 3

**Review:**

The paper introduces ModernGENA, a genomic foundation model that adapts recent advancements from ModernBERT to the domain of DNA sequence modeling. The authors argue that while recent DNA models have become increasingly complex, a modernized encoder-only baseline can achieve competitive performance with significantly higher computational efficiency. ModernGENA is benchmarked on the Nucleotide Transformer suite, where it ranks first among comparable encoder-only models and second overall against larger, multi-module systems.

Strengths:

S1. Strong Empirical Performance: ModernGENA achieves the top rank among the primary baseline set and ranks second overall even when compared to much larger models like GENERator.

S2. Computational Efficiency: The model demonstrates superior inference throughput, particularly when leveraging FlashAttention 2, and significantly outperforms established baselines across various sequence lengths.

S3. Practical Modernizations: The paper successfully integrates several state-of-the-art NLP architectural refinements into genomics, including unpadding, GeGLU, and a hybrid attention mechanism (alternating local and global attention).

S4. Reproducibility and Baseline Utility: By releasing the full implementation and pretrained weights, the authors provide a highly accessible and efficient baseline for the genomics community.

S5. Robust Training Strategy: The model was trained on a massive, diverse dataset using modern practices like dynamic sequence packing and a warmup-stable learning rate schedule.

Weaknesses

W1. Limited Context Length Evaluation: While ModernBERT is designed for long-context capability, the pretraining was conducted with a maximum sequence length of only 1024 tokens. The paper does not explicitly test the model's performance on extremely long-range genomic tasks that newer models are designed to handle.

W2. Incremental Architectural Innovation: The primary contribution is an adaptation of an existing NLP architecture (ModernBERT) rather than a novel architecture specifically designed for biological motifs.

W3. Modest Performance Gains: The authors themselves acknowledge that metric differences on the NT benchmark can be modest and highly sensitive to fine-tuning hyperparameters, making it difficult to definitively claim architectural superiority over smaller metric gaps.

W4. Lack of Task-Specific Biological Insight: The paper focuses heavily on technical benchmarks but provides limited discussion on why these specific transformer refinements are biologically more suitable for DNA compared to standard BERT or CNN-based models.

The paper provides a useful baseline for DNA foundation models. Its primary value lies in showing that efficient, well-engineered encoder-only transformers remain highly competitive against more complex contemporary systems. For a workshop focused on foundational models for science, this rigorous benchmarking and the release of a high-performance, accessible baseline are of significant value. I recommend a clear accept.

---

### Official Review · Reviewer_AAd4 · 2026-02-19
**ModernGENA as a Strong, Efficient Baseline for DNA Foundation Models**

**Rating:** 6
**Confidence:** 4

**Review:**

This paper presents ModernGENA, a DNA foundation model that adapts ModernBERT for genomics. Trained on promoter regions from 443 vertebrate genomes, it achieves competitive performance on the Nucleotide Transformer benchmark while maintaining higher inference throughput than comparable encoder-only models.

Strengths:
1) Rigorous efficiency benchmarking with clear methodology and practical insights.
2) Strong, reproducible baseline that the community needs.
3) Honest discussion of evaluation challenges in the field.
4) Demonstrates that careful engineering often outperforms naive scaling.

Weaknesses
1) Severe Training Data Bias: The promoter-centric pretraining needs clarification. While the model is evaluated on promoter-proximal tasks, the “foundation model” framing may overstate its generality.
2) Disconnect Between Context Claims and Training Reality: Long-context claims are insufficiently validated. The model is trained only up to 1,024 tokens but claims support for 8,192 without biological task validation.
3) Missing Ablation Studies: It is unclear which ModernBERT components drive improvements. The authors evaluate only the final composite model, limiting the paper’s utility as a guide for future architectural design.
4) Reproducibility Gap: The threshold for collapsing “N” runs is not specified.

These limitations are common in early DNA foundation model work and do not detract from the paper’s core value as an efficient, open baseline.

---

### Official Review · Reviewer_2mPb · 2026-02-23
**Review on Back to BERT in 2026**

**Rating:** 8
**Confidence:** 3

**Review:**

This manuscript introduces ModernGENA, a foundation model that adapts the recently proposed ModernBERT architecture to the domain of genomic sequence modeling. The authors pretrain two models—a base 135M parameter version and a large 377M parameter version—using a massive corpus of vertebrate promoter regions. By incorporating computational advancements such as rotary positional embeddings, hybrid attention, and optimized normalization, the work aims to provide a stable, long-context-capable baseline. The model is subsequently evaluated against existing architectures using the 18-task Nucleotide Transformer benchmark to assess its downstream transfer capabilities.

A major strength of this submission is its timely counter-narrative to the trend of increasingly complex, multi-module DNA language models. By focusing on rigorous architectural hygiene and efficiency, ModernGENA demonstrates an impressive performance-to-size ratio, ranking first among the primary baseline models under 500M parameters and second overall across all evaluated models. Furthermore, the authors provide thorough, controlled benchmarking of inference throughput, proving the model's superior processing speeds, particularly when utilizing FlashAttention 2. The explicit commitment to open science, including the release of the full implementation and pretrained weights, makes this a highly practical and reusable asset for the community.

While the authors astutely observe that absolute performance differences on the NT benchmark can be modest and heavily influenced by fine-tuning protocols, the evaluation still relies entirely on this single suite. Incorporating an orthogonal benchmark or a specific long-context task would more robustly validate the hybrid attention mechanism's capabilities.

It is a well-written, methodologically sound contribution. Instead of solely chasing scale, the authors prioritize computational efficiency and strong foundational baselines, yielding a modernized Transformer module that is both highly performant and accessible.

---

### Official Review · Reviewer_QML4 · 2026-02-23

**Rating:** 6
**Confidence:** 4

**Review:**

Pros:
- The paper presents a well-motivated study that carefully evaluates whether modern Transformer refinements transfer effectively to genomic modeling, which is a relevant and timely question given increasing model complexity in the field.
- The experimental methodology is rigorous, with controlled comparisons against strong encoder-only baselines and detailed reporting of efficiency metrics such as throughput across sequence lengths.
- The model demonstrates a strong efficiency–performance trade-off, achieving competitive benchmark results while improving inference throughput, which is valuable for practical deployment scenarios.


Cons:
- The conceptual novelty is limited because the work primarily adapts existing ModernBERT techniques rather than introducing new architectural or algorithmic ideas specific to genomics.
- The empirical evaluation focuses heavily on a single benchmark suite, which may not fully capture long-range genomic reasoning or real-world biological tasks.
- The reported performance improvements over baselines are generally modest, making it unclear whether the gains are practically meaningful beyond efficiency.
- The biological insights derived from the model are minimal, as the paper mainly emphasizes engineering improvements rather than scientific discovery.
- The comparison excludes some newer large-scale or non-Transformer architectures from the primary analysis, which may limit the completeness of the evaluation landscape.

---

### Decision · Program_Chairs · 2026-03-03

Accept (Poster)